# Using the Local Drought Data and GRACE/GRACE-FO Data to Characterize the Drought Events in Mainland China from 2002 to 2020

**Lilu Cui** [1,2] , **Cheng Zhang** [1] , **Zhicai Luo** [3,*] , **Xiaolong Wang** [4] , **Qiong Li** [5] **and Lulu Liu** [1]

1   School of Architecture and Civil Engineering, Chengdu University, Chengdu 610106, China;
    cuililu@cdu.edu.cn (L.C.); lilucui@whu.edu.cn (C.Z.); liululu@cdu.edu.cn (L.L.)
2   School of Geodesy and Geomatics, Wuhan University, Wuhan 430079, China
3   MOE Key Laboratory of Fundamental Physical Quantities Measurement & Hubei Key Laboratory of
    Gravitation and Quantum Physics, PGMF and School of Physics, Huazhong University of Science and
    Technology, Wuhan 430074, China
4   Nanning Survey and Design Institute Group Co., Ltd., Nanning 530022, China; wwxyzwxl@whu.edu.cn
5   School of Civil Engineering and Geomatics, Southwest Petroleum University, Chengdu 610500, China;
    qiongli@swpu.edu.cn
*   Correspondence: zcluo@hust.edu.cn; Tel.: +86-27-8754-3940

**Abstract:** Accurate quantification of drought characteristics helps to achieve an objective and comprehensive analysis of drought events and to achieve early warning of drought and disaster loss assessment. In our study, a drought characterization approach based on drought severity index derived from Gravity Recovery and Climate Experiment (GRACE) and its Follow-On (GRACE-FO) data was used to quantify drought characteristics. In order to improve drought detection capability, we used the local drought data as calibration criteria to improve the accuracy of the drought characterization approach to determine the onset of drought. Additionally, the local precipitation data was used to test drought severity determined by the calibrated drought characterization approach. Results show that the drought event probability of detection (*POD*) of this approach in the four study regions increased by 61.29%, 25%, 94.29%, and 66.86%, respectively, after calibration. We used the calibrated approach to detect the drought events in Mainland China (MC) during 2016 and 2019. The results show that CAR of the four study regions is 100.00%, 92.31%, 100.00%, and 100.00%. Additionally, the precipitation anomaly index (*PAI*) data was used to evaluate the severity of drought from 2002 to 2020 determined by the calibrated approach. The results indicate that both have a strong similar spatial distribution. Our analysis demonstrates that the proposed approach can serve a useful tool for drought monitoring and characterization.

**Keywords:** GRACE; GRACE-FO; drought characteristics; hydrological drought; TWSC

## 1. Introduction

Droughts are a serious natural phenomenon, which causes great damage to social life, agricultural production, economic development, and ecological environment [1,2]. As the global climate changes, the drought events occur frequently in many parts of the world, and their frequency and severity are gradually increasing [3]. The drought characteristics mainly include drought type, frequency, duration, severity, and drought area [4]. The quantification of drought characteristics has a very important role in disaster early warning, disaster loss reduction, and post-disaster reconstruction. The traditional drought characterization approach is based on the observation data (such as precipitation and evapotranspiration) derived from meteorological stations. However, this approach relies too much on the number and spatial distribution of sites, so it is difficult to obtain enough observation data to assess the drought characteristics in some areas where sites are scarce [5]. This approach has other disadvantages, such as high construction cost, difficulty

in obtaining large-scale and long-term observational data, and the overall situation of the terrestrial water [6].

The Gravity Recovery and Climate Experiment (GRACE) and its Follow-On (GRACE-FO) provides an alternative observation to monitor drought events from a new perspective [7,8]. It can detect the total terrestrial water storage changes (*TWSCs*) from the land surface to the deepest aquifers, such as surface storage, snow, soil moisture storage, groundwater storage, etc. [9]. Compared with the traditional approach, the GRACE and GRACE-FO data can reflect the situation of drought events from the perspective of the terrestrial water cycle well. In recent years, several new drought indices based on GRACE/GRACE-FO data were proposed to detect regional drought events. For example, Yirdaw et al. [10] used the GRACE-based total storage loss index (TSDI) to characterize the drought events in the Canadian Prairie River basin from 2002 to 2003, and verified them with local runoff data through the water balance equation. The results show that TSDI has the ability to detect drought events. Wang et al. [11] used the GRACE-based *TWSC* index (TWSI), precipitation anomaly index (PAI), and vegetation anomaly index (VAI) to analyze the drought situation in Haihe River basin of China from January 2003 to January 2013. The results indicate that TWSI is more suitable for monitoring long-term drought and severe drought in this region, because it can reflect the loss of deep soil water and groundwater compared with the other two drought indices. Yi and Wen [12] constructed a GRACE-based hydrological drought index (GHDI) to monitor the drought events in the United States from 2003 to 2012. The results show that the GHDI and Palmer Drought Severity Index (PDSI) have a good correlation. Sinha et al. [13] used GRACE *TWSC* data to construct the water storage deficit index (WSDI) and applied it to drought detection in India from April 2002 to April 2015, verifying the validity and reliability of the WSDI in quantifying large-scale drought events. Zhao et al. [14] used the GRACE-based drought severity index (GRACE-DSI) to detect drought events on a global scale from 2002 to 2014, and the results are highly consistent with the overall spatio-temporal distribution of PDSI and the standardized precipitation evapotranspiration index (SPEI). The above research results have proved that GRACE-based drought indices can be used for the detection of regional drought events.

However, the above studies were limited to discussing the drought detection ability of drought indices based on GRACE data. There are few studies on the characterization of drought characteristics (drought duration, drought severity, and drought area) using GRACE-based drought indices [15]. Thomas et al. [4] proposed a drought characterization approach based on water storage anomalies (WSAs) derived from GRACE data to detect and quantitatively characterize the drought events in the Amazon River basin, the Zambezi River basin, and southeastern USA and Texas. The results show that this approach can determine the duration of drought events and quantify the drought severity and peak magnitude. Sun et al. [5] used the above approach to characterize the drought characteristics in the Yangtze River basin from 2003 to 2015. The results show the Yangtze River Basin has experienced eight major drought events. Liu et al. [16] used a modified version of GRACE-DSI originally proposed by Zhao et al. [14] to characterize droughts in major watersheds in China during 2002–2017. This method has advantages, such as being a standardized index, allowing comparison of results between different locations. However, its ability to detect and characterize droughts has not been verified with an independent source of information. The main objective of this study was to evaluate the ability of GRACE-DSI to correctly detect and characterize drought episodes and to define an appropriate duration and severity thresholds for this drought index in four study regions in mainland China.

In our study, we first used the local drought data to verify the drought detection capability of the approach based on GRACE-DSI. Then, according to the verification results, the local drought data was used to calibrate the approach in order to improve the drought detection capability. Finally, we used the calibrated approach to characterize the drought events in Mainland China (MC) from 2002 to 2020. This paper is organized as follows. In Sections 2–4, we briefly introduce the study area, data, and methods used for analysis, respectively. Section 5 presents the evaluation results of the drought characterization

approach, parameter calibration process, and the results and the drought events were characterized by using the calibrated approach in MC from 2002 to 2020. The discussion and conclusion are provided in Sections 6 and 7, respectively.

## 2. Study Area

Since the MC has the characteristics of wide regional scope, large geographical environment differences, and various climates, we selected one region in each of the four directions in the east, south, west, and north to conduct the research on drought event characterization (see Figure 1).

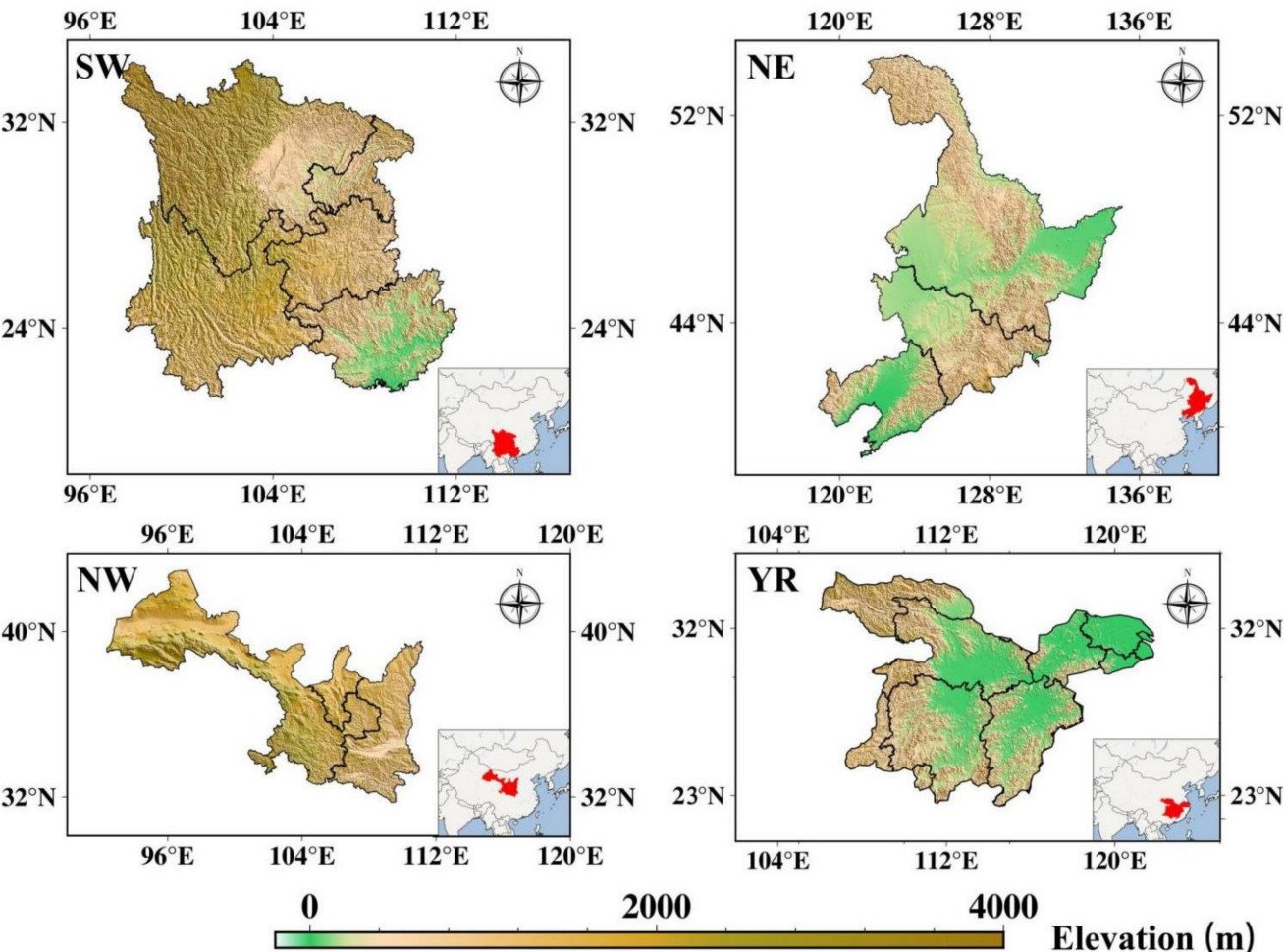

**Figure 1.** The digital elevation models of the study areas (SW: Five Southwest provinces; NE: Three Northeast Province; NW: Three Northwest province; YR: The middle and lower reaches of Yangtze River basin).

These four regions in Figure 1 are five southwest provinces (SW, including Sichuan, Chongqing, Yunnan, Guizhou, and Guangxi), three northeast provinces (NE, including Hei longjing, Ji lin, and Liao ning), three northwest provinces (NW, including Shanxi, Gansu, and Ningxia), and the middle and lower reaches of Yangtze River (YR, including Hubei, Hunan, Jiangxi, Shanghai, and parts of Shanxi, Anhui, Jiangsu, Zhejiang, and Guizhou), respectively.

## 3. Data

### 3.1. GRACE/GRACE-FO Data

The GRACE and GRACE-FO RL06 monthly Spherical Harmonic (SH) products (truncated to degree and order 60) were used to calculate local TWSC in accordance with the approach presented by Wahr et al. [17]. All the SH products used in our study are published by Center for Space Research (CSR) at the University of Texas at Austin, including GRACE

SH products from April 2002 to June 2017 and GRACE-FO SH products covering the period June 2018 to July 2020.

Before inversion calculation, the SH products need to be preprocessed. Firstly, make the geocentric correction to its 1-degree coefficients [18]; secondly, C20 coefficients were replaced by the ones from Satellite Laser Ranging (SLR) [19]; thirdly, we used the combination filter processing consisting of a 300 km Fan filter [20] and de-correlation filter P3M6 [21] to reduce the impact of high frequency and correlated errors. Finally, the inversion results used the scale factor method to restore the signals weakened by the degree truncation and filter processing [22]. Through the above four steps of data processing, the grid data of *TWSC* in the study region can be obtained. Since GRACE and GRACE-FO data are essentially the same, we collectively referred to GRACE and GRACE-FO data as GRACE data in this study.

### 3.2. Reconstructed TWSC Data

As the GRACE and GRACE-FO missions are not coherent, there is an 11-month data gap between the two [23]. In this paper, we used the dataset of reconstructed terrestrial water storage in China based on precipitation (2002–2019) to fill this gap, which was provided by National Tibetan Plateau Data Center. This dataset was calculated from multiple sources, including CSR GRACE/GRACE-FO RL06 Mascon Solutions, China's daily gridded precipitation real-time analysis system (version 1.0), and CN05.1 temperature data and other data sets by using a precipitation reconstruction model, and considering the seasonal items and trend item of CSR RL06 Mascon products [24]. The data set contains 191 months of monthly *TWSC* data, of which GRACE data is 161 months, GRACE-FO data is 17 months, and the GRACE and GRACE-FO intermittent periods are 11 months. The calculation formula of reconstruction *TWSC* data is as follows [25]:

$$TWSC_{rec} = \beta \cdot P^{\tau} \tag{1}$$

where $TWSC_{rec}$ is the reconstruction TWSC data, $P$ is monthly precipitation data, $\beta$ is the calibration parameter of the long-term trend term, and $\tau$ is the calibration parameter of the seasonal term. To evaluate the quality of the data set, we calculated the RMS of GRACE/GRACE-FO *TWSC* data and reconstructed *TWSC* data in the same time period. The specific results is presented in the Appendix A.

### 3.3. Standardized Precipitation Evapotranspiration Index (SPEI) Data

The SPEI is calculated from the cumulative sum of the difference between precipitation and potential ET on different time scales following the same procedure to compute the standardized precipitation index (SPI) [26]. In this study, the spatial resolution of SPEI gridded data is $1° \times 1°$. These datasets were calculated by using the precipitation data derived from Global Precipitation Climatology Centre and the potential ET data derived from National Oceanic and Atmospheric Administration. SPEI has three time scales of 3 months, 6 months, and 12 months. The 3-month scale is associated with variations in soil moisture: agricultural drought; the 6-month scale is associated with variations in streamflow: hydrological drought; and the 12-month scale is associated with variations in groundwater storage: hydrogeological drought [27]. Therefore, SPEI data on different time scales can be used for monitoring different types of drought events. In this study, we used these gridded data from April 2002 to December 2018.

### 3.4. Self-Calibrating Palmer Drought Severity Index (SCPDSI) Data

The SCPDSI is a variant on the original PDSI of Palmer [28], with the aim of making results from different climate regimes more comparable. As with the PDSI, the SCPDSI was calculated from the time series of precipitation and temperature, together with fixed parameters related to the soil/surface characteristics at each location [29,30]. The monthly global gridded SCPDSI data with a spatial resolution $0.5° \times 0.5°$ were provided by Climate

Research Unit (CRU) at University of East Anglia. In this study, we used these gridded data from April 2002 to December 2019.

### 3.5. In Situ Drought Data

The in situ meteorological drought data was derived from the Chinese Flood and Drought Yearbooks that were published by the China Ministry of Water Resource [31] for the period of 2006–2019. In the yearbooks, the definition of a drought disaster is an event that causes water shortages due to reduced precipitation and insufficient water supply for water projects and causes harm to life, production, and ecology. The yearbooks provided the start and end time of drought disaster, the administrative region affected by drought, the loss of crops, the number of people who have difficulty accessing drinking water, and a textual description of drought disaster. In our study, we used the information of the start and end time of the drought disaster to evaluate the accuracy of the approach used in this paper to detect the drought events.

### 3.6. In Situ Precipitation Data

Monthly gridded precipitation data in MC for the time period between April 2002 and December 2020 provided by the China National Meteorological Science Data Center and sorted by the National Meteorological Information Center with a spatial resolution $0.5° \times 0.5°$ were used for the analysis.

## 4. Methods

### 4.1. GRACE-DSI

Before calculating GRACE-DSI, we needed to detrend the time series of GRACE/GRACE-FO-based TWSC, that is, subtract the long-term trend signal from the original time series. Then, the detrended TWSC time series was standardized to obtain GRACE-DSI. The formula is as follows [14]:

$$GRACE - DSI_{i,j} = \frac{TWSC_{i,j} - TWSC_j^{mean}}{\sigma_j} \tag{2}$$

where $TWSC_{i,j}$ is the TWSC in the $i$ year (ranging from 2002 to 2020) $j$ month (ranging from January to December), calculated according to the Warh et al. [18]. $TWSC_j^{mean}$ and $\sigma_j$ are the average and standard deviation of $TWSC$ in month $j$, respectively. Table 1 presents the drought severity categories according to the GRACE-DSI value [14,32].

**Table 1.** GRACE-DSI drought grades classification.

| Type | GRACE-DSI |
|---|---|
| Exceptional Drought | $\leq -2.0$ |
| Extreme Drought | $-2.0 - -1.6$ |
| Severe Drought | $-1.6 - -1.3$ |
| Moderate Drought | $-1.3 - -0.8$ |
| Light Drought | $-0.8 - -0.5$ |
| No Drought | $> -0.5$ |

### 4.2. Precipitation Anomaly INDEX (PAI)

PAI is used for comparison with GRACE-DSI in drought detection. The specific expression is as follows [11]:

$$PAI = \frac{PPT_{i,j} - PPT_j^{mean}}{PPT_j^{mean}} \tag{3}$$

where $PPT_{i,j}$ is the precipitation in the $i$ year (ranging from 2002 to 2020) and $j$ month (ranging from January to December) and $PPT_j^{mean}$ is the average value of TWSC in the $j$ month.

### 4.3. Drought Characterization

According to the research results of Liu et al. [16], a drought episode is defined as being when the GRACE-DSI is less than $-0.8$ for three consecutive months. When the drought episode appears in 20% of the area of this region, a regional drought event has occurred. Notably, when the interval between two adjacent drought events is only one month and the GRACE-DSI in this month is less than $-0.8$ in any fraction of the region, the two adjacent drought events are merged into one drought event; otherwise, there are two independent drought events [16,33].

The duration of the drought is the time elapsed from the moment when the above conditions begin to be met until the moment when they are no longer met, i.e., when at least 80% of the area of the region has a GRACE-DSI value greater than $-0.8$. The drought-affected area indicates the area of the grid where the drought event occurred. The drought magnitude represents the maximum value of GRACE-DSI during a drought event. The expression of drought severity ($S$) is as follows [4]:

$$S = \overline{M} \times D \tag{4}$$

where $\overline{M}$ is the monthly average of GRACE-DSI during the drought event and $D$ is the drought duration both calculated up to the current month.

### 4.4. Evaluation Metrics

To test the monitoring capabilities of the drought characterization approach based on GRACE-DSI, the classification of hit, miss, false alarm, and correct negative were applied [34] (Table 2).

**Table 2.** Contingency table for drought detection.

|  | **Drought Occurred** | **No Drought Occurred** |
|---|---|---|
| Drought detected | Hit | False alarm |
| No drought detected | Miss | Correct negative |

For this evaluation, two indices were used: the probability of detection (*POD*) and the false alarm rate (*FAR*). The expression of *POD* is as follows:

$$POD = \frac{a}{a+b} \tag{5}$$

where $a$ is the number of hits and $b$ is the number of misses. The expression of *FAR* is as follows:

$$FAR = \frac{c}{c+d} \tag{6}$$

where $c$ is the number of false alarms, $d$ is the number of correct negatives. The values of both indices range from 0 to 1, with 1 being the optimal value for CAR, while 0 is the optimal value for *FAR*.

### 4.5. Parameter Calibration Approach

In our study, we used the local drought data to calibrate the parameters of the drought characterization approach as follows:

(1)　The CAR and FAR of drought characterization approach were calculated considering the initial set of spatial and severity thresholds described in Section 4.3;

(2) Severity and spatial thresholds were modified to achieve the detection of 90 % of the months that were actually found to be under drought conditions based on local drought data;

(3) According to the threshold determined in step 2, *POD* and *FAR* of GRACE-DSI were recalculated.

(4) If *POD* and *FAR* were improved compared to the previous ones, then *POD* and *FAR* were increased or decreased by 0.1 on the basis of the threshold set in step 3. Then, *POD* and *FAR* were recalculated again. Otherwise, it means that the threshold set in step 3 are most suitable.

## 5. Results

### 5.1. The Drought Characterization Approach Based on GRACE-DSI

Figure 2 shows the time series of GRACE-DSI, SCPDSI, SPEI-3, SPEI-6, and SPEI-12 in MC from April 2002 to July 2020. It can be seen that the variation of the five drought indices time series is basically the same. It initially verifies the reliability of drought monitoring by GRACE-DSI in the above four regions. In order to further verify the above results, the correlation coefficients between these drought indices are shown in Figure 3. It can be seen that GRACE-DSI is strongly correlated with the other four drought indices, except for NW. The correlation coefficients between GRACE-DSI and SCPDSI are the highest in the above study regions, and the highest value (0.73) appears in SW. The weakest correlation with GRACE-DSI is SPEI-3, and the lowest value of the correlation coefficient between GRACE-DSI and SPEI-3 appeared in YR (0.51). We found a weak correlation between GRACE-DSI and the other four drought indices in NW, which is consistent with the results of Liu et al. [17]. It may be related to the local climate characteristics in this region, because SCPDSI and SPEI are mainly based on the local precipitation and evapotranspiration data, while GRACE-DSI is based on TWSC. However, the precipitation in NW accounts for a much smaller proportion of terrestrial water storage than those of the other three regions [32]. The difference between GRACE-DSI and the other drought indices was magnified in this region. Therefore, the results of region NW show the reliability of GRACE-DSI in drought monitoring. Overall, it is reliable to use GRACE-DSI for drought monitoring in the above four regions.

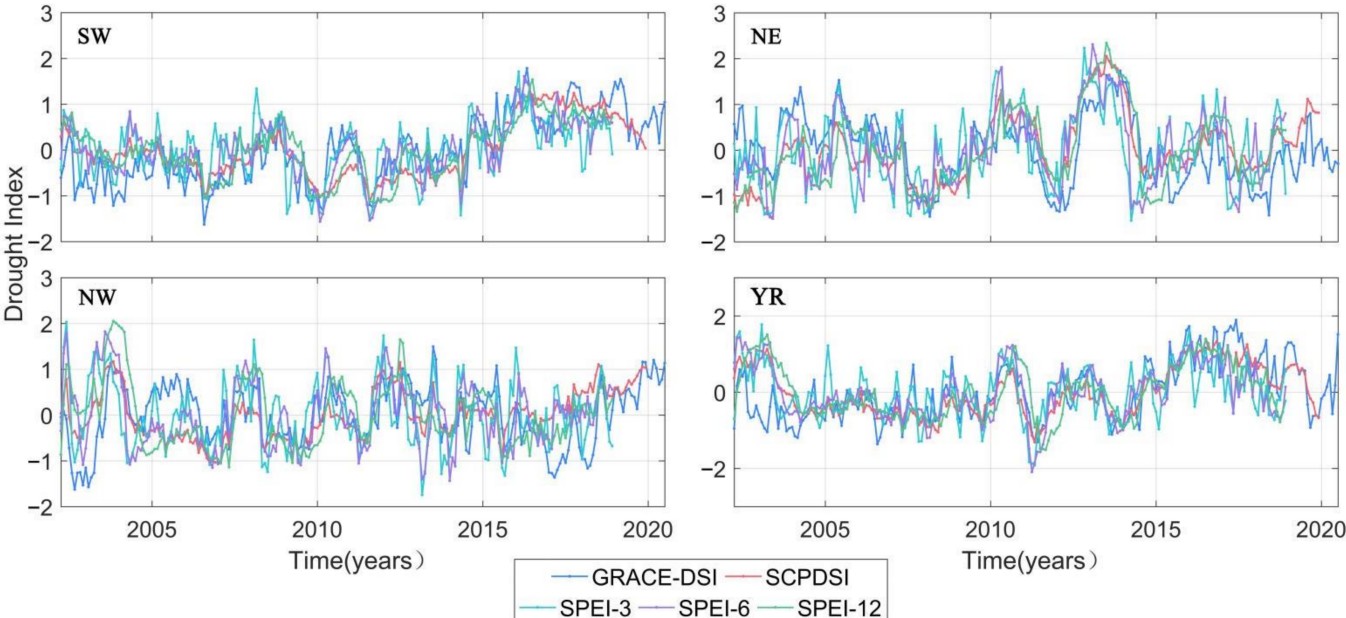

**Figure 2.** The time series of GRACE-DSI, SCPDSI, SPEI-3, SPEI-6, and SPEI-12 in MC during April 2002 and July 2020.

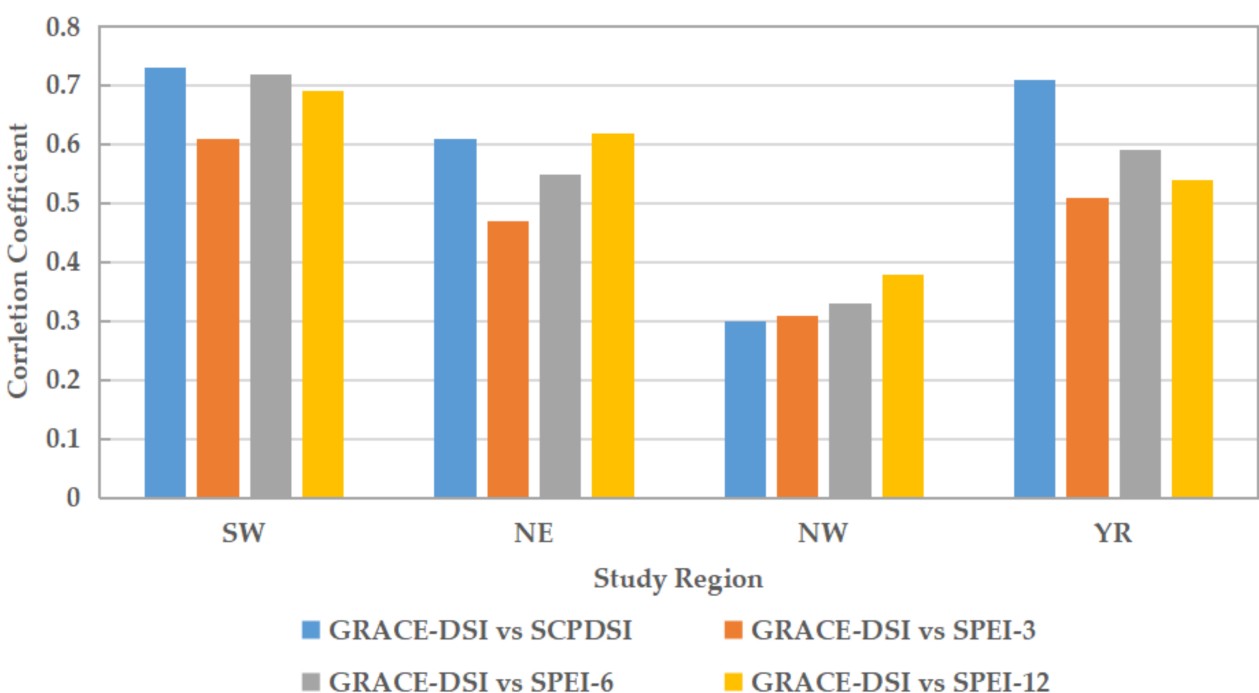

**Figure 3.** The correlation coefficient between GRACE-DSI and SCPDSI, SPEI-3, SPEI-6, and SPEI-12.

According to the definition of Section 4.2, the drought events were detected from April 2002 to July 2020 in the four study regions, the characteristics of which were quantitatively analyzed (Figure 4 and Table 3). From Figure 4 and Table 3, we found that during the study period, a total of five drought events occurred in SW. The most severe and longest drought event appeared from September 2009 to March 2010, which lasted 7 months. The drought severity of this event was −7.43 and the average drought area percentage was 0.75. The peak value of GRACE-DSI (−1.30) appeared in March 2010, and the drought area percentage was 0.81. The least influential drought event occurred from February 2003 to April 2003, which lasted 3 months. The drought severity was −2.82, and the average drought area percentage was 0.47. In NE, there were four drought events. The most influential one appeared during September 2011 and July 2012, lasting as long as 11 months, and the drought severity reached −12.19. On average, 69% of the regions were affected by drought. The GRACE-DSI reached the peak value (−1.34) in February 2012, and the drought area percentage was 0.74. The least one occurred from June to October 2015, whose duration was 5 months. The corresponding drought severity and drought area percentage were −4.97 and 0.67, respectively. The three drought events occurred in NW. The most severely affected occurred from August 2002 to April 2003 (9 months in total). The drought severity was −12.28 and more than 69% of the regions were affected by drought. The peak value of GRACE-DSI was −1.62 in September 2002, while at the same time, the drought area percentage was 0.75. YR experienced three drought events. The most severe one occurred from April to September 2011. The drought severity was −6.51 and the drought events appeared in more than 78% of the regions. There was a peak value of GRACE-DSI (−1.31) in June 2011 and the drought area percentage was 0.96 in this month.

In summary, the drought characterization approach based on GRACE-DSI can realize the monitoring of drought events, and it can also quantify drought duration, drought magnitude, drought area, and drought severity. However, it requires deeper discussion and analysis of the reliability and accuracy of drought event detection by using this approach.

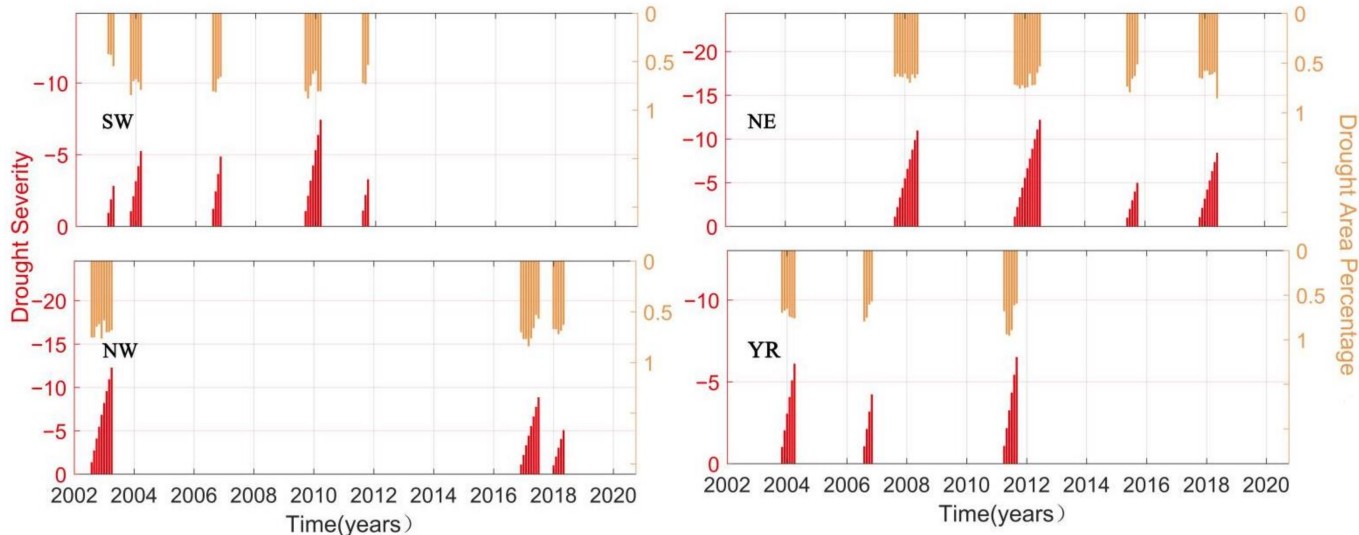

**Figure 4.** Drought events (drought duration, drought severity, and drought-affected area) in Mainland China during April 2002 and July 2020 (the red bars represent the drought severity, orange bars represent the drought area percentage).

**Table 3.** Summary table of drought events identified by GRACE-DSI.

| Region | No. of Drought Events | Time Span of Drought Events | Duration (months) | GRACE-DSI | | | Drought Area Percentage | | |
| --- | --- | --- | --- | --- | --- | --- | --- | --- | --- |
| | | | | Peak Magnitude | Average Magnitude | Severity | Peak Magnitude | Average Magnitude | Cumulative Manitude |
| SW | 5 | 200302–200304 | 3 | −1.14 | −0.94 | −2.82 | 55.02% | 47.15% | 140.89% |
| | | 200311–200403 | 5 | −1.19 | −1.02 | −5.24 | 85.41% | 75.50% | 373.60% |
| | | 200608–200611 | 4 | −1.63 | −1.22 | −4.86 | 81.89% | 74.11% | 296.02% |
| | | 200909–201003 | 7 | −1.30 | −1.06 | −7.43 | 88.70% | 75.04% | 527.30% |
| | | 201108–201110 | 3 | −1.24 | −1.09 | −3.28 | 73.42% | 66.74% | 199.95% |
| NE | 4 | 200709–200806 | 10 | −1.45 | −1.10 | −10.96 | 70.00% | 64.22% | 637.65% |
| | | 201109–201207 | 11 | −1.34 | −1.11 | −12.19 | 76.30% | 69.52% | 761.22% |
| | | 201506–201510 | 5 | −1.04 | −0.99 | −4.97 | 80.00% | 67.50% | 334.20% |
| | | 201711–201806 | 8 | −1.42 | −1.05 | −8.40 | 86.12% | 64.25% | 515.22% |
| NW | 3 | 200208–200304 | 9 | −1.62 | −1.36 | −12.28 | 77.72% | 69.66% | 622.32% |
| | | 201612–201707 | 8 | −1.36 | −1.11 | −8.85 | 84.24% | 70.05% | 561.30% |
| | | 201801–201805 | 5 | −1.16 | −1.01 | −5.05 | 72.40% | 68.66% | 338.85% |
| YR | 3 | 200311–200404 | 6 | −0.88 | −1.02 | −6.11 | 76.81% | 71.09% | 428.81% |
| | | 200608–200611 | 4 | −1.36 | −1.06 | −4.23 | 80.05% | 68.47% | 273.29% |
| | | 201104–201109 | 6 | −1.31 | −1.05 | −6.51 | 96.60% | 78.10% | 468.90% |

### 5.2. The Calibration of the Drought Characterization Approach

The results of local drought data from 2006 to 2015 were compared with the ones of the drought characterization approach. The results are shown in Figure 5 and Table 4. It can be seen that the performance of total *FAR* (3.32%) is good, but the one of total *POD* (24.83%) is relatively poor. Therefore, this approach has room for improvement. The results of the four regions are not the same. The highest *POD* (62.50%) appears in NE, but the *FAR* is only 11.46%. The regions where the FAR is 0 occur in SW, NW, and YR, but the *POD* in SW and YR are only 22.58% and 28.57%, respectively. Unfortunately, the *POD* in NW is 0.00%. According to the above results, this approach requires certain measures to be improved to ensure that *POD* is as high as possible and *FAR* is as low as possible.

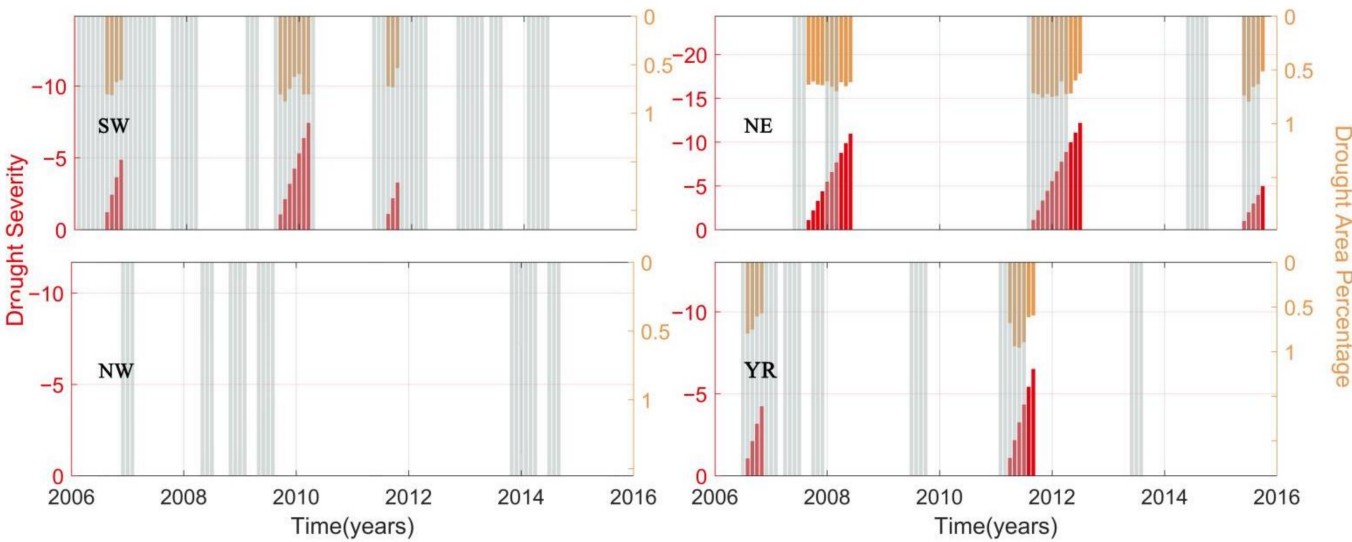

**Figure 5.** The comparison of the drought events determined by the local meteorological drought data and the drought characterization approach based on GRACE-DSI during 2006~2015 (the red bars represent the drought severity, orange bars represent the drought area percentage, and the gray bars represent the drought duration determined by the local meteorological drought data).

**Table 4.** The *POD* and *FAR* of the drought characterization approach in four different regions.

| Region | POD | FAR |
|---|---|---|
| SW | 22.58% | 0.00% |
| NE | 62.50% | 11.46% |
| NW | 0.00% | 0.00% |
| YR | 28.57% | 0.00% |
| Total | 24.83% | 3.32% |

In order to maximize the effect of the improved approach, we counted the situation of monthly GRACE-DSI and drought area percentage in the four study regions during the drought events derived from the local drought data. The results are shown in Table 5. Considering that a drought is defined when the GRACE-DSI is less than $-0.5$, the areas with a grid value less than $-0.5$ are considered to be in drought, and the grid area is included in the statistics of the drought area. It can be seen from Table 5 that the probability that the monthly GRACE-DSI is below $-0.8$ is only 34.68%, so it is unreasonable that the threshold of monthly GRACE-DSI was set to $-0.8$. The threshold for defining the occurrence of a drought proposed by Zhao (2017; Table 1) is $-0.5$; however, the probability that the monthly GRACE-DSI is below $-0.5$ is only 66.13%. In order to ensure that the most drought events are detected, the detection thresholds need to be cover at least 90% of the drought events (90% criterion adopted was arbitrarily proposed by the authors). We found that the probability that the monthly GRACE-DSI is below $-0.2$ is 89.52%, and the one that the monthly GRACE-DSI is below $-0.1$ is 95.16%. The statistical results of the drought area percentage show that the probability is greater than 90% only when the drought area percentage is less than 10% (94.70%). Therefore, we initially set the threshold of monthly GRACE-DSI and the percentage of drought area to $-0.2$ and 10%. Subsequently, we determined the most suitable threshold according to the procedure in Section 4.5.

**Table 5.** Frequency table of the drought area percentage and monthly GRACE-DSI in four study regions.

| | Drought Area Percentage | | | Monthly GRACE-DSI | |
| --- | --- | --- | --- | --- | --- |
| Data Range | Frequency | Percentage | Data Range | Frequency | Percentage |
| 0–10% | 7 | 5.30% | 0–−0.1 | 6 | 4.84% |
| 10%–20% | 11 | 8.33% | −0.1–−0.2 | 7 | 5.65% |
| 20%–30% | 8 | 6.06% | −0.2–−0.3 | 9 | 7.26% |
| 30%–40% | 9 | 6.82% | −0.3–−0.4 | 9 | 7.26% |
| 40%–50% | 11 | 8.33% | −0.4–−0.5 | 11 | 8.87% |
| 50%–60% | 10 | 7.58% | −0.5–−0.6 | 12 | 9.68% |
| 60%–70% | 17 | 12.88% | −0.6–−0.7 | 12 | 9.68% |
| 70%–80% | 21 | 15.91% | −0.7–−0.8 | 15 | 12.10% |
| 80%–100% | 38 | 28.79% | −0.8–−∞ | 43 | 34.68% |

Figure 6 and Table 6 show the corresponding results of the calibrated drought characterization approach. The threshold of monthly GRACE-DSI and drought area percentage of SW and NE are −0.1 and 10%, respectively, while the ones of NW and YR are −0.2 and 10%, respectively. Comparing Tables 4 and 6, we found that after the local drought data calibration processing, total FAR increased by 20.2% and the increase in total *POD* was 65.69%. Among them, the *POD* of the four study regions (SW, NE, NW, and YR) is 83.87%, 87.50%, 94.29%, and 96.43%, respectively, while the *FAR* of the four study regions is 12.07%, 32.29%, 24.71%, and 25.00%, respectively. Although *FAR* increased, the increase of *POD* is much greater than that of *FAR*. Overall, the effect of restraint is more obvious.

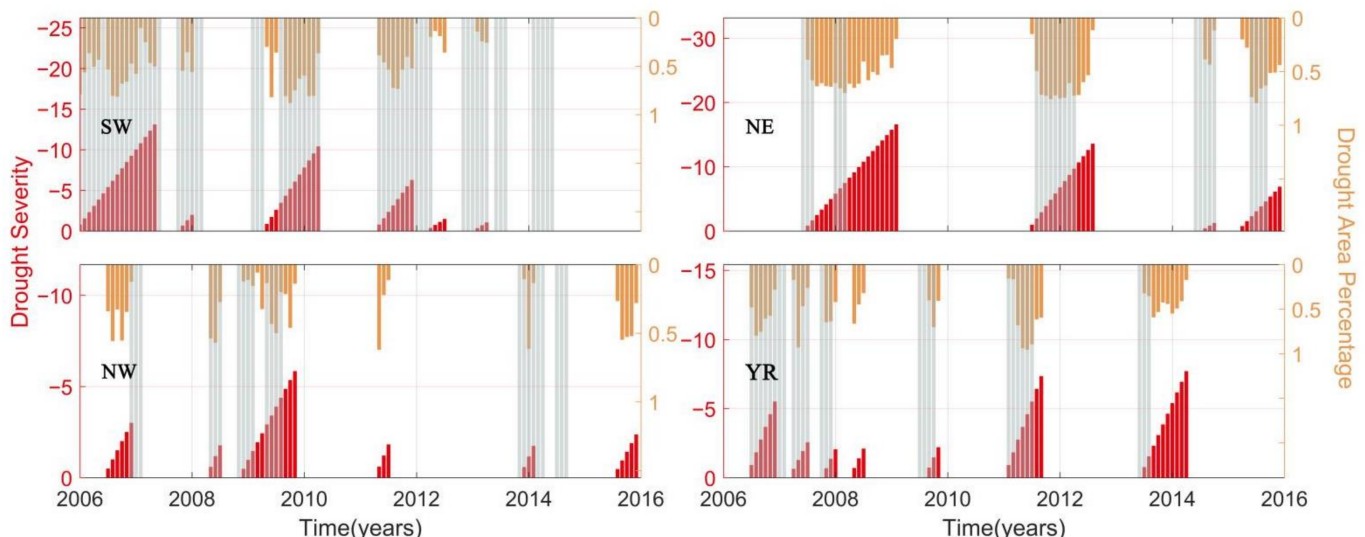

**Figure 6.** The drought characteristics determined by the calibrated drought characterization approach during 2006~2015.

**Table 6.** The POD and FAR of the drought event characterization approach in four different regions during 2006 and 2015.

| Region | *POD* | *FAR* |
| --- | --- | --- |
| SW | 83.87% | 12.07% |
| NE | 87.50% | 32.29% |
| NW | 94.29% | 24.71% |
| YR | 96.43% | 25.00% |
| Total | 90.52% | 23.52% |

In order to verify the effect of the calibration effect, we calculated the results of *POD* and *FAR* of the calibrated approach from 2016 to 2019. The results are shown in Table 7 and Figure 7. According to the results, we found that after improvement, the total *POD* and *FAR* of the drought characterization approach in the four study regions are 98.08% and 19.67%. Among them, the *POD* and *FAR* of SW and YR are the same, which are 100.00% and 0.00%, respectively. At the same time, the *POD* of NE and NW is also very high, at 92.31% and 100.00%, respectively. However, the drought characterization approach has a certain *FAR* in these two regions, which are 40.91% and 37.78%. Through the above analysis, it can be seen that the improvement measures in this paper have a significant effect on the accuracy of the drought characterization approach.

**Table 7.** The *POD* and *FAR* of the calibrated approach in four different regions during 2016 and 2019.

| Region | POD | FAR |
|--------|--------|--------|
| SW | 100.00% | 0.00% |
| NE | 92.31% | 40.91% |
| NW | 100.00% | 37.78% |
| YR | 100.00% | 0.00% |
| Total | 98.08% | 19.67% |

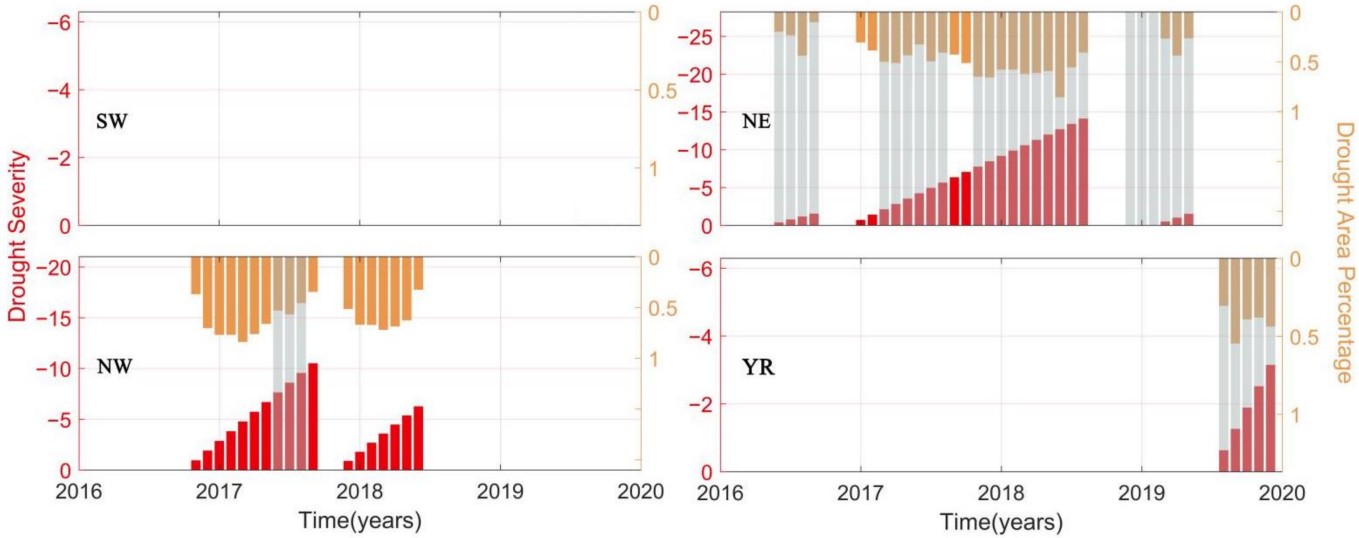

**Figure 7.** The comparison of the drought events determined by the calibrated drought characterization approach during 2016~2019.

### 5.3. Characterization of Drought Events

We used the calibrated drought characterization approach to detect drought events that occurred in the four study regions from April 2002 to July 2020, which are shown in Figure 8 and Table 8.

From Figure 8 and Table 8, it can be seen that the number of drought events in the four study regions (SW, NE, NW, and YR) from 2002 to 2020 are 8, 8, 9, and 10, respectively. During this period of time, the time of severe drought events in SW are from September 2002 to June 2004 and from June 2005 to May 2007, the ones in NE are from July 2007 to February 2009 and January 2017 to August 2018, the ones in NW are from July 2002 to July 2003 and from March 2016 to June 2018, and the ones in YR are from September 2002 to July 2004 and from July 2013 to April 2014.

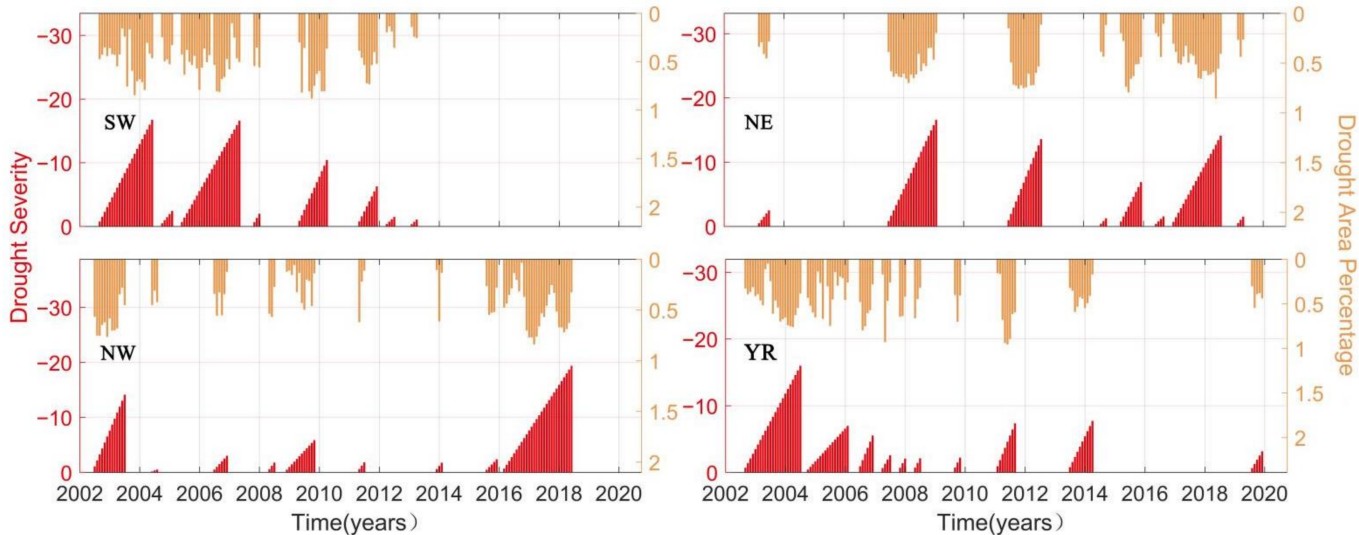

**Figure 8.** Drought events (drought duration, drought severity, and drought-affected area) determined by the calibrated approach in Mainland China during April 2002 and July 2020.

**Table 8.** Summary table of drought events identified by the calibrated approach.

| Region | No. of Drought Events | Time Span of Drought Events | Duration (months) | GRACE-DSI | | | Drought Area Percentage | | |
|---|---|---|---|---|---|---|---|---|---|
| | | | | Peak Magnitude | Average Magnitude | Severity | Peak Magnitude | Average Magnitude | Cumulative Manitude |
| SW | 8 | 200209–200406 | 22 | −1.21 | −0.76 | −16.73 | 91.49% | 62.84% | 1382.56% |
| | | 200410–200502 | 5 | −0.64 | −0.48 | −2.41 | 68.96% | 62.31% | 311.56% |
| | | 200506–200705 | 24 | −1.63 | −0.69 | −16.60 | 95.41% | 71.32% | 1711.71% |
| | | 200711–200801 | 3 | −0.74 | −0.66 | −1.98 | 78.81% | 70.82% | 212.47% |
| | | 200905–201004 | 12 | −1.30 | −0.87 | −10.41 | 93.90% | 72.16% | 865.88% |
| | | 201105–201112 | 8 | −1.24 | −0.78 | −6.28 | 83.03% | 69.91% | 559.28% |
| | | 201204–201207 | 4 | −0.51 | −0.37 | −1.49 | 48.42% | 41.04% | 164.17% |
| | | 201302–201304 | 3 | −0.48 | −0.35 | −1.06 | 39.85% | 33.17% | 99.51% |
| NE | 8 | 200303–200307 | 5 | −0.64 | −0.50 | −2.49 | 55.30% | 46.34% | 231.68% |
| | | 200707–200902 | 20 | −1.45 | −0.83 | −16.58 | 74.14% | 62.34% | 1246.80% |
| | | 201107–201208 | 14 | −1.34 | −0.97 | −13.58 | 83.27% | 72.86% | 1020.07% |
| | | 201408–201410 | 3 | −0.66 | −0.42 | −1.25 | 58.33% | 47.48% | 142.43% |
| | | 201504–201512 | 9 | −1.14 | −0.77 | −6.89 | 94.02% | 68.47% | 616.22% |
| | | 201606–201609 | 4 | −0.72 | −0.39 | −1.54 | 70.37% | 45.97% | 183.88% |
| | | 201701–201808 | 20 | −1.42 | −0.71 | −14.12 | 91.46% | 60.10% | 1201.93% |
| | | 201903–201905 | 3 | −0.68 | −0.51 | −1.52 | 68.64% | 60.12% | 180.35% |
| NW | 9 | 200207–200307 | 13 | −1.62 | −1.09 | −14.11 | 83.59% | 69.42% | 902.43% |
| | | 200406–200408 | 3 | −0.21 | −0.17 | −0.51 | 50.11% | 47.75% | 143.26% |
| | | 200607–200612 | 6 | −0.76 | −0.50 | −3.01 | 70.49% | 58.56% | 351.36% |
| | | 200805–200807 | 3 | −0.80 | −0.59 | −1.77 | 71.60% | 63.71% | 191.14% |
| | | 200812–200911 | 12 | −0.75 | −0.49 | −5.84 | 82.17% | 50.46% | 605.50% |
| | | 201105–201107 | 3 | −0.83 | −0.61 | −1.82 | 79.14% | 52.01% | 156.04% |
| | | 201312–201402 | 3 | −0.90 | −0.58 | −1.74 | 89.56% | 48.80% | 146.40% |
| | | 201508–201512 | 5 | −0.74 | −0.47 | −2.37 | 62.83% | 54.54% | 272.68% |
| | | 201603–201806 | 28 | −1.36 | −0.69 | −19.37 | 88.95% | 60.71% | 1699.92% |
| YR | 10 | 200209–200407 | 23 | −1.19 | −0.69 | −15.97 | 79.35% | 58.49% | 1345.22% |
| | | 200410–200602 | 17 | −0.88 | −0.41 | −6.97 | 83.38% | 55.30% | 940.07% |
| | | 200607–200612 | 6 | −1.36 | −0.92 | −5.51 | 88.40% | 80.11% | 480.67% |
| | | 200704–200707 | 4 | −1.17 | −0.64 | −2.56 | 97.58% | 72.15% | 288.58% |
| | | 200711–200801 | 3 | −0.79 | −0.69 | −2.06 | 75.09% | 70.46% | 211.39% |
| | | 200805–200807 | 3 | −0.76 | −0.70 | −2.10 | 83.58% | 76.67% | 230.02% |
| | | 200909–200911 | 3 | −0.90 | −0.74 | −2.21 | 92.34% | 72.70% | 218.10% |
| | | 201102–201109 | 8 | −1.31 | −0.92 | −7.35 | 99.51% | 76.58% | 612.61% |
| | | 201307–201404 | 10 | −1.11 | −0.77 | −7.71 | 93.28% | 64.71% | 647.15% |
| | | 201908–201912 | 5 | −0.93 | −0.63 | −3.15 | 66.04% | 53.98% | 269.89% |

In order to verify the drought severity determined by this approach, we used the PAI severity derived from local precipitation data to compare the drought severity derived from GRACE-DSI. Due to the many drought events, it is inconvenient to show them one by one. Only the results of the two most severe drought events in each region are shown in Figure 9. From Figure 9, we can see that the spatial distribution of the GRACE-DSI and *PAI* severity

in the regions NE and NW have good spatial similarity, while the similarity between the two data is relatively weak in the regions SW and YR. It may be related to the main factors affecting the occurrence of drought events in SW and YR, not only precipitation but also runoff [32]. Additionally, there are also differences in the spatial distribution of the two data in the four study regions. It is related to the different hydrological components reflected by the two data. GRACE *TWSC* data represent the total change of various water inputs and outputs that cause *TWSC*. However, precipitation is only one of the water inputs that cause *TWSC*.

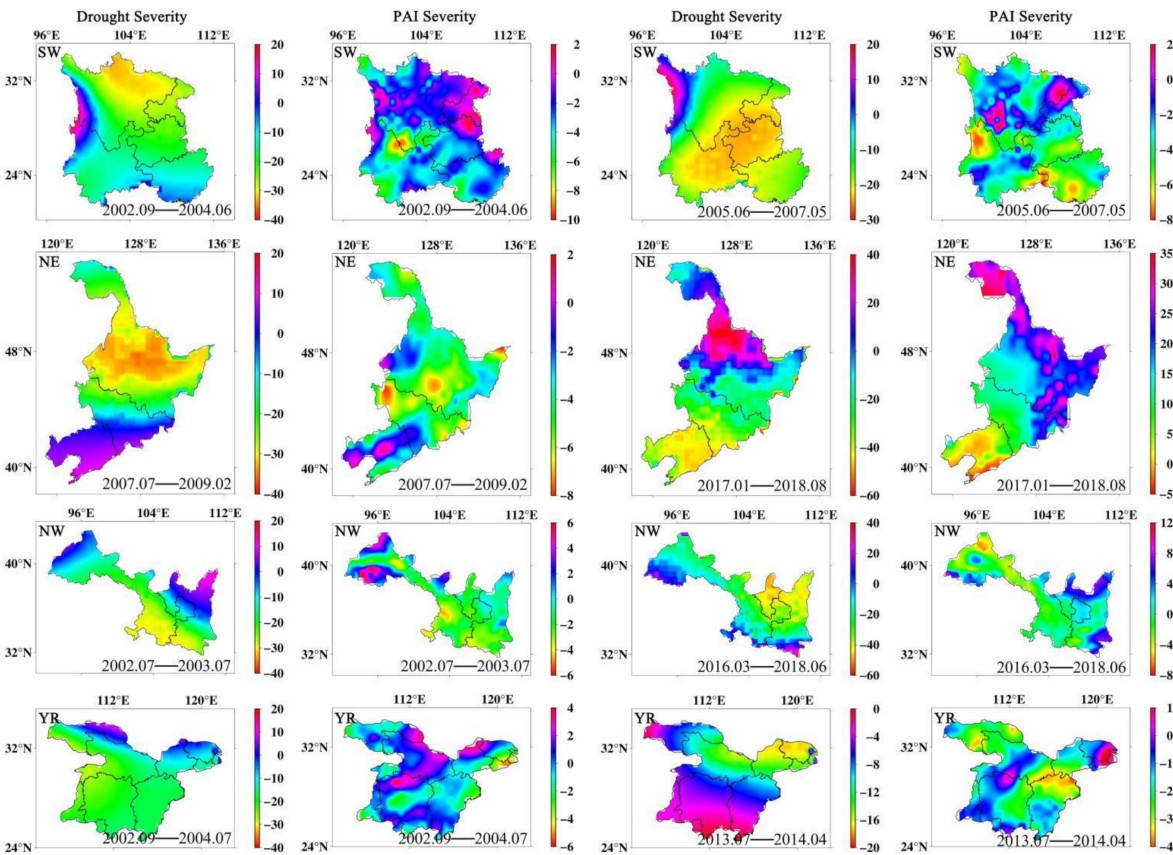

**Figure 9.** Spatial distribution plot of drought severity and *PAI* severity of drought events in four study regions.

## 6. Discussion

The GRACE mission provides considerable insight for drought monitoring and characterization. In addition to detecting the deficit of surface water storage and shallow groundwater storage, it can also detect the deficit of deep groundwater storage [14]. Therefore, the drought index based on GRACE data can reflect the influence of drought on deep groundwater storage change. We can see that there are strong correlations between GRACE-DSI and SCPDSI and SPEI in the regions studied, with the exception of NW. However, there are differences between them, which may be related to the difference in their algorithm and hydrological components. GRACE-DSI represents a collection of multiple hydrological components in a certain region, including the water input and output [35], but PDSI and SPEI only consider one or a few water inputs and outputs, such as precipitation and evapotranspiration [13,16]. SPEI only considers precipitation and temperature, while PDSI is calculated by using the potential values of a certain variable, which may lead to an exaggeration of drought conditions [5,13]. In addition to the above meteorological factors, GRACE-DSI also considers the influence of human activities on *TWSC*. Therefore, it can be said that GRACE-DSI is a comprehensive drought index. Compared with the traditional drought index, it has significant advantages in the drought monitoring and assessment.

According to the results of Table 7, the *FAR* of NE and NW are 40.91% and 37.78%, respectively. We analyze the causes of false detection in the two regions separately. In NW, there is actually only one drought event that was falsely detected. The false detected drought only lasted 8 months, which meant that the *FAR* is only 17.78%. The remaining false detected drought months are due to the drought event detected by GRACE-DSI lasting longer than the one identified by the local drought meteorological data. Additionally, in NE, there is no false detected drought event. The *FAR* is mainly caused by the inconsistency in the duration of drought events determined by the kinds of data. It may be caused by the inconsistency in the definition of drought events between the two data. The GRACE-DSI is used mainly to detect the hydrological drought, while the local meteorological drought data is more concerned with the impact of drought on agriculture and social economy [5]. Although the results show that the most suitable detection thresholds of different regions are different, but because the research regions involved in this paper are less, it is not enough to support the above results. Further, it is beyond the scope of this paper.

Our research results show that the drought characterization approach based on GRACE-DSI can monitor and characterize the regional drought events. Additionally, it proves that using local drought data to calibrate the drought characterization approach is an effective means to improve the drought detection capability of the approach. However, the local drought data in this paper only provides the start and end time of drought events and lacks other drought characteristic data. Therefore, obtaining more detailed local drought data is an important follow-up work of our study.

## 7. Conclusions

In this paper, we combined the local drought data and GRACE *TWSC* data to construct a drought characterization approach, and used this approach to monitor and characterize the drought events in MC from April 2002 to July 2020. *PAI* was used to verify the drought severity derived from GRACE-DSI. The results show that after the local drought data was calibrated, the total *POD* of the drought characterization approach increased by 65.69%. We also used the local drought data from 2016 to 2019 to evaluate the *POD* of the calibrated approach. The results show that the *POD* reached 98.08%. Finally, by comparing *PAI*, we found that the drought severity determined by the calibrated approach has a certain degree of reliability.

Our research results have certain reference significance for accurately quantifying drought characteristics, and also help to realize early warning of drought events and provide strong scientific support for relevant government departments to formulate drought resistance and disaster reduction policies.

**Author Contributions:** Conceptualization, L.C. and Q.L.; methodology, L.C.; software, C.Z.; validation, L.C. and C.Z.; resources, L.L.; data curation, L.C.; writing—original draft preparation, L.C.; writing—review and editing, L.C., X.W. and Q.L.; visualization, L.C.; supervision, Z.L.; funding acquisition, Z.L. All authors have read and agreed to the published version of the manuscript.

**Funding:** This research was funded by the National Key R&D Program of China (Grant No. 2018YFC1503503), the National Natural Science Foundation of China (Grant No. 41931074, 42061134007), the project funded by the Key Laboratory of Geospace Environment and Geodesy, Ministry of Education, Wuhan University (Grant No. 16-01-04, 18-02-04).

**Institutional Review Board Statement:** Not applicable.

**Informed Consent Statement:** Not applicable.

**Data Availability Statement:** GRACE/GRACE-FO RL06 data: http://icgem.gfz-potsdam.de/series (accessed on 16 September 2021); Reconstructed TWSC data: (http://data.tpdc.ac.cn); SPEI data: (http://spei.csic.es/); SCPDSI data: (https://crudata.uea.ac.uk/cru/data/drought/ (accessed on 16 September 2021); In-situ drought data: http://www.mwr.gov.cn/sj/tjgb/zgshzhgb/ (accessed on 16 September 2021); In-situ precipitation data: http://data.cma.cn (accessed on 16 September 2021).

**Acknowledgments:** We are grateful to the Center of Space Research (CSR) for providing the monthly GRACE gravity field solutions, and to the Goddard Space Flight Center for providing the monthly GLDAS-2.1 data, to the National Tibetan Plateau Data for providing the Dataset of reconstructed terrestrial water storage in China based on precipitation (2002–2019), to the China National Meteorological Science Data Center for providing the monthly precipitation products, to the Climatic Research Unit at university of East Anglia for providing the SCPDSI data, to Vicente-Serrano S.M. and Beguería S for providing the SPEI database and to the China Ministry of Water Resource for providing the Chinese Flood and Drought Yearbooks.

**Conflicts of Interest:** The authors declare no conflict of interest.

## Appendix A

In order to verify the quality of reconstructed TWSC data, we used the generalized three-cornered hat method [36,37] to calculate the root mean square (RMS) of GRACE/GRACE-FO TWSC data and reconstructed TWSC data, respectively. Since there is an 11-month data gap between GRACE and GRACE-FO, we compared the RMS of reconstructed TWSC data with the ones of GRACE-based and GRACE-FO-based TWSC data, respectively. The results are shown in Table A1, Figures A1 and A2. From these two figures, we can see that the RMSs of reconstructed TWSC data are greater than the ones of GRACE/GRACE-FO-based TWSC data. This is because the hydrological information reflected by these two types of data are quite different. GRACE/GRACE-FO-based TWSC data reflects the TWSC caused by climate factors and human activities, while reconstructed TWSC data is calculated based on precipitation data, and the precipitation data is only one of the climate factors that caused TWSC.

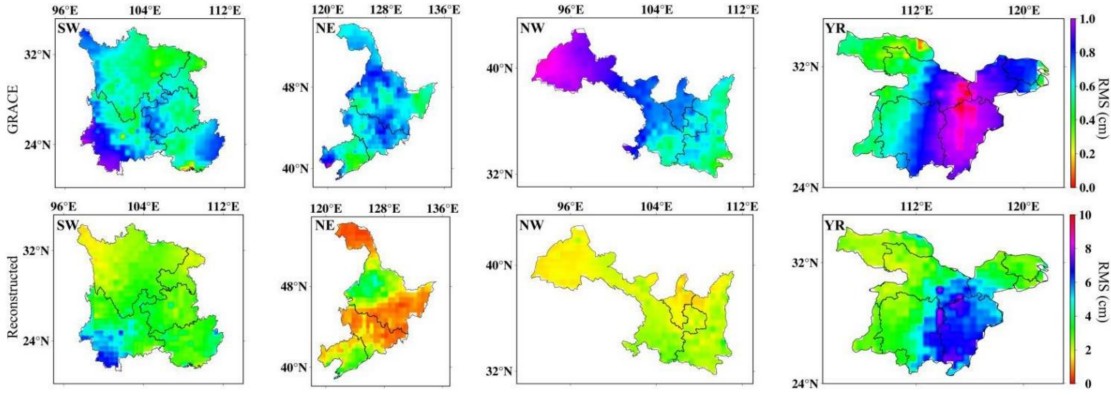

**Figure A1.** RMS plots of GRACE-based TWSC and reconstructed TWSC in the four study regions from April 2002 to June 2017.

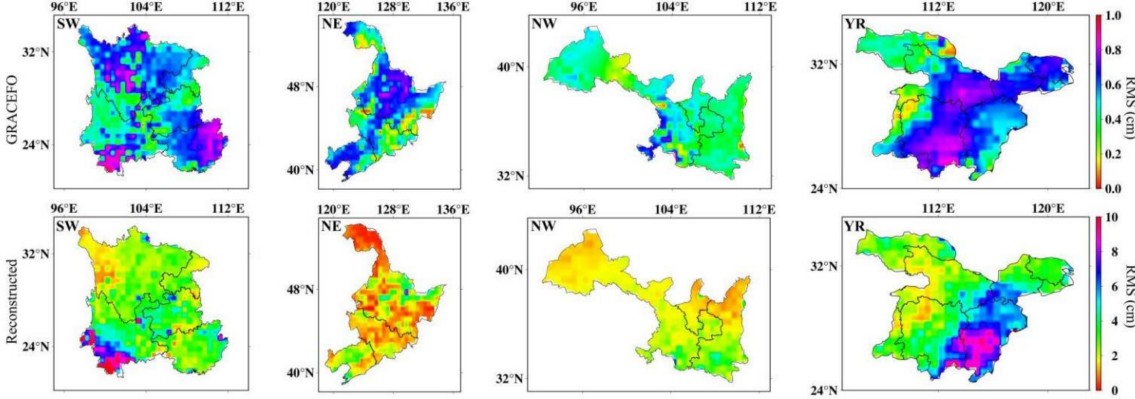

**Figure A2.** RMS plots of GRACE-FO-based TWSC and reconstructed TWSC in the four study regions from June 2018 to July 2020.

**Table A1.** RMS of GRACE/GRACE-FO-based TWSC and reconstructed TWSC data (unit: cm).

| Data Type | SW | NE | NW | YR |
|---|---|---|---|---|
| GRACE-based TWSC | 0.61 | 0.65 | 0.59 | 0.74 |
| Reconstructed TWSC (from April 2002 to June 2017) | 3.12 | 2.48 | 1.94 | 4.10 |
| GRACE-FO-based TWSC | 0.46 | 0.50 | 0.42 | 0.58 |
| Reconstructed TWSC (from June 2018 to June 2020) | 3.33 | 2.14 | 1.88 | 4.17 |

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
