# Peer review of "Using the Local Drought Data and GRACE/GRACE-FO Data to Characterize the Drought Events in Mainland China from 2002 to 2020"

_applsci, doi:10.3390/app11209594_

Round 1

Reviewer 1 Report

Please find my comments attached

Author Response

The response to Reviewer 1 is in the attachment

Reviewer 2 Report

Comments in the attached PDF file.

Round 2

Reviewer 1 Report

The authors have addressed all my concerns. I am happy to recommend this paper to publish with some minor revisions.

Section 3. The authors need to explain why they use PPT from different sources for reconstructed TWSC and SPEI.

L224 RAR -> CAR

L317-319. Need to rephrase this paragraph for clarification.

Author Response

Response to Review 1 is in the attachment

Reviewer 2 Report

Please, see the attached document.

Author Response

Response to Reviewer 2 is in the attachment
